# Absence of CCR2 Promotes Proliferation of Alveolar Macrophages That Control Lung Inflammation in Acute Respiratory Distress Syndrome in Mice

**DOI:** 10.3390/ijms232112920

**Published:** 2022-10-26

**Authors:** Vivian Louise Soares de Oliveira, Emilie Pollenus, Nele Berghmans, Celso Martins Queiroz-Junior, Marfa Blanter, Matheus Silvério Mattos, Mauro Martins Teixeira, Paul Proost, Philippe E. Van den Steen, Flávio Almeida Amaral, Sofie Struyf

**Affiliations:** 1Laboratory of Immunopharmacology, Department of Biochemistry and Immunology, Institute of Biological Sciences, Federal University of Minas Gerais, Belo Horizonte 31270-901, MG, Brazil; 2Laboratory of Molecular Immunology, Department of Microbiology, Immunology and Transplantation, Rega Institute for Medical Research, KU Leuven, 3000 Leuven, Belgium; 3Laboratory of Immunoparasitology, Department of Microbiology, Immunology and Transplantation, Rega Institute for Medical Research, KU Leuven, 3000 Leuven, Belgium; 4Laboratory of Immunopharmacology, Department of Morphology, Institute of Biological Sciences, Federal University of Minas Gerais, Belo Horizonte 31270-901, MG, Brazil

**Keywords:** ARDS, lung inflammation, CCR2, chemokine, resolution of inflammation, monocytes, immunology

## Abstract

Acute respiratory distress syndrome (ARDS) consists of uncontrolled inflammation that causes hypoxemia and reduced lung compliance. Since it is a complex process, not all details have been elucidated yet. In a well-controlled experimental murine model of lipopolysaccharide (LPS)-induced ARDS, the activity and viability of macrophages and neutrophils dictate the beginning and end phases of lung inflammation. C-C chemokine receptor type 2 (CCR2) is a critical chemokine receptor that mediates monocyte/macrophage activation and recruitment to the tissues. Here, we used CCR2-deficient mice to explore mechanisms that control lung inflammation in LPS-induced ARDS. CCR2^−/−^ mice presented higher total numbers of pulmonary leukocytes at the peak of inflammation as compared to CCR2^+/+^ mice, mainly by enhanced influx of neutrophils, whereas we observed two to six-fold lower monocyte or interstitial macrophage numbers in the CCR2^−/−^. Nevertheless, the time needed to control the inflammation was comparable between CCR2^+/+^ and CCR2^−/−^. Interestingly, CCR2^−/−^ mice presented higher numbers and increased proliferative rates of alveolar macrophages from day 3, with a more pronounced M2 profile, associated with transforming growth factor (TGF)-β and C-C chemokine ligand (CCL)22 production, decreased *inducible nitric oxide synthase (Nos2)*, *interleukin (IL)-1β* and *IL-12b* mRNA expression and increased *mannose receptor type 1 (Mrc1)* mRNA and CD206 protein expression. Depletion of alveolar macrophages significantly delayed recovery from the inflammatory insult. Thus, our work shows that the lower number of infiltrating monocytes in CCR2^−/−^ is partially compensated by increased proliferation of resident alveolar macrophages during the inflammation control of experimental ARDS.

## 1. Introduction

Acute respiratory distress syndrome (ARDS) was first described in 1967 [1] and is defined as noncardiogenic pulmonary edema leading to a respiratory failure with diffuse bilateral pulmonary infiltrate and tissue injury, besides severe hypoxemia [2]. The pathogenesis of ARDS includes the dysfunction of the alveolar-capillary membrane, leading to excessive transendothelial and transepithelial leukocyte migration and the influx of protein-rich edema fluid into the alveolar space. The inflammation is worsened by the release of several pro-inflammatory mediators that can also be cytotoxic, increasing the destruction of the membrane and diffuse tissue damage [3,4,5]. ARDS is caused by pulmonary or systemic inflammation following gastric aspiration, pneumonia, COVID-19, sepsis and trauma [6]. Due to the diverse causes and complex pathogenesis, ARDS treatment is also unspecific, poorly described, and considered an important unmet medical need [7,8]. In addition to the high rates of morbidity and mortality, ARDS has a great impact on the quality of life of patients requiring a better understanding of their pathogenesis and new treatment options [9,10].

The acute inflammatory response consists of an intricate but well-coordinated chain of actions that involves molecular, cellular, and physiological changes [11]. The recognition of the initial insults by lung resident cells causes the production and release of a plethora of mediators that trigger several inflammatory events. Among the cells involved in the different phases of inflammation, the alveolar macrophages (AM) are crucial. Being the most abundant innate immune cell in the alveolar spaces of the lungs [12], AM are the first line of defense against infections and invaders, recognizing pathogen-associated molecular patterns, such as LPS from Gram-negative bacteria. They are able to phagocytose and eliminate these pathogens and release pro-inflammatory cytokines to induce immune cell recruitment and the development of the inflammation [13]. Additionally, AM are very important in the late stages of ARDS since the depletion of these cells has been linked with decreased efferocytosis and lowered control of inflammation [14,15].

The release of several chemotactic factors leads to a broad recruitment of leukocytes to the lung parenchyma and alveolar space, including polymorphonuclear (PMNs) and mononuclear cells. CCR2 is an important chemokine receptor that plays a fundamental role in monocyte recruitment and activation by the recognition of its high-affinity ligand CCL2 [16,17]. Initially, the early accumulation of monocytes, monocyte-derived macrophages, and PMNs in the lungs determines local inflammation. Moreover, the activation state and viability of these cells modulate the different phases of inflammation, from the beginning to its resolution. The resolution of inflammation is essential to restore the tissue to its physiological functioning after the damage caused by the foreign insult and the inflammatory response. Impairment of this process may lead to an unresolved inflammation, which lies beneath the pathogenesis of several chronic inflammatory disease processes [18]. While recruited CCR2^+^ monocytes have a crucial role in the onset of inflammation, their presence in the tissue together with the recruitment of non-phlogistic monocytes in later phases helps to control the inflammation. An important event that causes the shift to the resolution of inflammation is the apoptosis of PMNs and their subsequent engulfment by local macrophages. This phenomenon is called efferocytosis and drives the differentiation of macrophages and their polarization into a pro-resolving profile, stimulating the production and release of pro-resolving mediators that suppress the progression of inflammation and promote tissue repair [19,20,21]. 

Various experimental models have been used to investigate the molecular mechanisms of ARDS, with LPS-induced ARDS as one of the most common models [22]. An advantage of this model is the possibility to investigate the mechanisms inherent to the different phases of lung inflammation, from the early events to its resolution and tissue repair [23]. Here, we explored in this model the impact on lung inflammation of the CCL2–CCR2 axis through the use of CCR2 knock-out mice, both at the early pro-inflammatory phase and during the resolution of inflammation.

## 2. Results

### 2.1. Lack of CCR2 Modifies the Recruitment Profile of Monocytes and Neutrophils in Early Time Points after LPS Instillation

Before studying the role of CCR2 in the model of LPS-induced ARDS, we evaluated the levels of its ligand, CCL2, in the bronchoalveolar lavage fluid (BALF) of CCR2^+/+^ and CCR2^−/−^ mice and observed increased levels of this chemokine mainly on days 2 and 3 after the insult, but at remarkedly higher levels in CCR2^−/−^ mice when compared to CCR2^+/+^ mice (Figure 1A). Next, we analyzed the differences in inflammatory profile in lungs in both mice upon intranasal LPS challenge. 

Regarding the cell accumulation, leukocyte numbers increased on days 1 to 3 after LPS challenge, with higher total cell numbers in CCR2^−/−^ mice, mostly neutrophils (Figure 1B,C). In contrast, CCR2^+/+^ presented with a higher accumulation of macrophages derived from monocytes in the first two time points as compared to CCR2-deficient mice (Figure 1D). Interestingly, despite differences in the profile of cells accumulated in the lung at days 1 to 3, both strains had comparable numbers of cells at later time points, when the cell counts returned to the basal levels (from day four onwards). In order to evaluate the impact of those differences in cell influx on lung pathology, we analyzed the total protein concentration in BALF to quantify pulmonary edema, and the changes in body weight. However, there were no differences in those parameters between CCR2^+/+^ and CCR2^−/−^ mice during the whole period evaluated (Figure 1E,F), despite being clear that on the first 3 days after the challenge, both CCR2^+/+^ and CCR2^−/−^ mice had protein leakage into the alveolar space. Thus, we investigated other parameters to understand the impact of the differing leukocyte profile in the lungs on tissue inflammation.

### 2.2. Cytokine Production in the Initial Phases of Inflammation Is Altered in the Absence of CCR2, but Does Not Impact the Tissue Damage

Cytokines and chemokines were measured in BALF to better determine the inflammatory profile of this ARDS model in CCR2^+/+^ and CCR2^−/−^ mice. IFN-γ and TNF-α are important cytokines associated with tissue inflammation and damage caused by LPS. Both cytokines are increased in CCR2^+/+^ mice, at day 2 after LPS insult for both and at day 3 only for IFN-γ (Figure 2A,B). 

Of note, no increase in those cytokines was measured in CCR2-deficient mice at any of the time points evaluated. However, the level of CXCL1, an important chemokine related to neutrophil recruitment was increased in CCR2^−/−^ mice already at day 1 (Figure 2C), which can explain the more pronounced accumulation of neutrophils in CCR2^−/−^ mice (Figure 1C) when compared to CCR2^+/+^ mice. Consequently, more neutrophil gelatinase-associated lipocalin (NGAL), a protein released specifically by activated neutrophils, was observed already very early in the absence of CCR2 (Figure 2D).

Despite the outspoken difference in cell accumulation and cytokine production, no significant alterations were detected on histology. Compared to healthy mice, both CCR2^+/+^ and CCR2^−/−^ mice presented higher histopathological scores at day 2 after LPS instillation, as observed in Figure 3. At day 5 after the challenge, the histopathological score is reduced for both mice, being comparable with the healthy control groups. Interestingly, CCR2^+/+^ and CCR2^−/−^ have similar results at every time point evaluated, suggesting that, despite the differences previously demonstrated at the peak of inflammation, the inflammatory response is resolved within the same time frame in both strains.

### 2.3. The Profile of Monocytes/Macrophages Varies between CCR2^+/+^ and CCR2^−/−^ Mice 

CCR2 is an important receptor for monocyte recruitment in the early stages of tissue inflammation. The accumulation of these cells in lung tissue directly contributes to increased inflammation, but the recruited cells also contribute to the end stages of inflammation, with a crucial participation at the resolution of inflammation and in tissue repair [24]. Thus, we evaluated the profile of monocytes and macrophages at different time points after LPS-induced ARDS. 

As expected, the absence of CCR2 prevented vast accumulation of macrophages (CD45^+^CD11b^+^Ly6G^−^SiglecF^−^CD3^−^NKp46^−^CD19^−^CD103^−^), inflammatory monocytes (CD45^+^CD11b^+^SiglecF^−^Ly6G^−^CD3^−^NKp46^−^CD19^−^CD103^−^CD64^+^Ly6C^+^), and interstitial macrophages (CD45^+^CD11b^+^SiglecF^−^Ly6G^−^CD3^−^NKp46^−^CD19^−^CD103^−^CD64^+^MHCII^+^) in CCR2^−/−^ mice when compared to CCR2^+/+^ mice at days 1 to 3 after the challenge (Figure 1D and Figure 4A,B). In contrast, the number of alveolar macrophages (CD45^+^SiglecF^+^CD11c^+^) was significantly higher at days 3 and 4 in the CCR2-deficient mice compared to the CCR2^+/+^ mice, which rather maintained the same alveolar macrophage counts along the whole duration of the experiment (Figure 4C).

Since there was a significant increase in the number of alveolar macrophages in the CCR2-deficient mice, the proliferation of these cells was evaluated. Two different assays were performed: the analysis of Ki-67 expression (Figure 5A,B) and the assessment of BrdU incorporation in the DNA (Figure 5C,D). Interestingly, the expression of Ki-67 in the alveolar macrophages was enhanced, and more alveolar macrophages expressed Ki-67 at 3 days after LPS challenge. This effect was even more pronounced in the CCR2-deficient mice. In the CCR2^−/−^ group, more BrdU had been incorporated at 3 and 4 days after the LPS challenge in the alveolar macrophages. These results indicate that the lack of CCR2 is linked to the increase in alveolar macrophage proliferation on days 3 and 4, timepoints associated with the reduction of neutrophils and most likely the beginning of resolution of inflammation.

### 2.4. Alveolar Macrophages Can Be Associated with the Final Events of Tissue Inflammation and Its Resolution in the Absence of CCR2

Different parameters are associated with the resolving phase of acute inflammation, such as the polarization of macrophages to an M2 profile and the production of pro-resolving mediators. Analysis of the expression of CD206, a marker indicative of M2-polarization in macrophages, showed that the numbers of CD206^+^ alveolar macrophages were enhanced after 3 days in CCR2^+/+^ and CCR2^−/−^ mice. Mice deficient for CCR2 had even more alveolar macrophages expressing CD206 at day 4 after the LPS stimulation compared to CCR2^+/+^ mice (Figure 6A). In contrast, alveolar macrophages expressing NOS2, a marker for M1 polarization of macrophages, were decreased in the absence of CCR2 (Figure 6B). Confirming these results, the ratio of *Nos2* over *Arginase 1 (Arg1)* mRNA expression was significantly reduced in the lungs of mice deficient for CCR2 (Figure 6C). In addition, pulmonary *IL-1β*, and *IL-12* expression was reduced, while *Mrc1* mRNA was increased in CCR2^−/−^ mice (Appendix A). At day 3, the deficiency of CCR2 also led to increased protein levels of TGF-β and CCL22 compared to the wild type mice (Figure 6D,E). TGF-β is an important cytokine related to resolution of inflammation that is able to induce apoptosis of leukocytes [25]. Both TGF-β and CCL22 are differentially produced by M2 macrophages [26,27]. Therefore, the absence of CCR2 is associated with the increase of AM expressing CD206, the increase of other M2 markers in the lungs/BALF (*Mrc1*, CCL22 and TGF-*β*) and a reduction of M1 markers (*Nos2*, *IL-1β*, *IL-12b*).

### 2.5. Depletion of Alveolar Macrophages before the LPS Challenge Leads to Uncontrolled Inflammation Which Is Worsened in the Absence of CCR2

To further demonstrate the role of alveolar macrophages in the absence of CCR2, CCR2^+/+^ and CCR2^−/−^ mice were treated with clodronate-loaded liposomes. As observed in Figure 7A,B, the depletion of alveolar macrophages was successful since the percentage and absolute numbers of this specific cell population were reduced. Depletion triggered an increase in the number of total leukocytes and neutrophils in the alveolar space in both CCR2^+/+^ and CCR2^−/−^ mice at 4 days after the LPS challenge (Figure 7C,D). Interestingly, more leukocytes were detected in CCR2^−/−^ compared to CCR2^+/+^ mice. To evaluate the impact of alveolar macrophage depletion on lung pathology, we analyzed the total protein concentration in the BALF to quantify pulmonary edema. Figure 7E shows that both CCR2^+/+^ and CCR2^−/−^ mice had more pulmonary edema after the depletion, but that the inflammatory insult had still more impact in CCR2 KO mice on day 4, probably because the resolution of inflammation is delayed in those mice. Lastly, we evaluated the changes in bodyweight and, while its reduction was observed in all the groups during the course of the inflammation, only the mice treated with clodronate-loaded liposomes were not able to recover and still weighed significantly less at day 4 (Figure 7F).

## 3. Discussion

The resolution of lung inflammation requires an orchestrated immune response and several control mechanisms to avoid excessive inflammation and chronic disease [28,29]. CCR2 is a crucial receptor that regulates tissue inflammation through its fundamental role in monocyte recruitment. The CCL2-CCR2 axis plays an important role in monocyte biology, guiding the compartmentalization of these cells in different tissues during homeostasis and inflammation. CCR2 deficient mice are known to have lower numbers of circulating Ly6C^Hi^ cells, since CCR2 is required for the mobilization of monocytes from the bone marrow to the circulation during a systemic inflammatory response [30]. It has been demonstrated that CCR2 is important in the development of inflammation in lungs (asthma [31], tuberculosis [32] and pulmonary fibrosis [33]), liver [34], myocardium [35,36] and others [37] due to its importance in monocyte recruitment.

In this study, we used CCR2-deficient mice to understand the kinetics of lung inflammation using an experimental model of ARDS induced by LPS, which can elicit a powerful pro-inflammatory, though self-resolving immune response [38]. The lack of CCR2 generally leads to a decrease of monocytes/macrophages at the site of the inflammation, which may lead to a milder disease [35,39]. In contrast with our findings, Maus et al. [40,41] and Francis et al. [42] showed that the absence or blocking of CCR2 dramatically reduced the recruitment of myeloid cells in general, and not only the monocyte/macrophage population, impacting the disease parameters greatly in models of ARDS induced by LPS and ozone, respectively. Similarly, depletion of circulating monocytes by intravenous injection of clodronate liposomes 2 days before intratracheal LPS treatment significantly suppressed the acute lung injury in mice [43]. Adversely, our data show that the reduced monocyte influx does not prevent development of inflammation in the model of ARDS induced by intranasal low-dose LPS instillation. We found that in the initial phases of the inflammation, the absence of CCR2 led to a dramatic decrease in the accumulation of macrophages in the lungs and an increase in the recruitment of neutrophils, congruous with the higher levels of CXCL1 in the BALF. Contrastingly, at later time points we did not observe major differences in the body weight kinetics, inflammatory parameters or immunopathological score between the two mouse strains indicating that although lack of CCR2 does not prevent lung inflammation, it does not hamper adequate resolution. We discovered that absence of CCR2 was compensated by increased proliferation of alveolar macrophages that were more skewed towards an M2 phenotype as we detected an increased expression of the M2 marker CD206 on alveolar macrophages, and higher levels of CCL22 and TGF-β in the BALF. In addition, pulmonary *Nos2*, *IL-1β*, and *IL-12b* expression was reduced, while *Mrc1* was increased in CCR2^−/−^ mice (Appendix A and Figure 6). Interestingly, the lower expression of *IL-12b* might be connected with the reduced levels of IFN-γ observed in CCR2-deficient mice (Figure 2) [44] and, consequently, the reduction of NOS2 [45]. Together, those elements are indicative for efficient resolution of inflammation in the CCR2-deficient mice as the general paradigm states that resolution of acute inflammation is characterized by the accumulation of pro-resolving macrophages that phagocytose apoptotic cells and produce pro-resolving molecules [46].

The effect of CCR2 absence at later time points of inflammation is indeed ambivalent. Previous reports showed that the lack of CCR2 signaling (a) reduces pro-fibrotic responses in the lungs [34,39]; (b) refrains extracellular matrix remodeling [34], (c) delays the resolution of inflammation and the recovery of the gastrointestinal functions [43]; (d) improves cardiac remodeling [36], and (e) limits recovery following spinal cord injury [47]. In our study, the deficiency of CCR2 did not change the resolution timeline, suggesting that this receptor is not crucial in this acute and self-resolving model of lung inflammation. This is in agreement with the study by Pollenus et al. [48], who observed that CCR2 is dispensable for the resolution of malaria-induced lung pathology. Together, these studies indicate that CCR2 divergently affects the development of different diseases probably depending on the organ involved, the profile and timing of each aspect in the inflammatory response. According to the mice, the model, and the type of inflammation, monocytes/macrophages may be beneficial for the proper development and resolution of inflammation, and they may impact other leukocytes differentially. 

Even though CCR2 is essential for the recruitment of monocytes, in the absence of this receptor, a minor increase in monocyte derived macrophages, monocytes, and interstitial macrophages at 2 and 3 days after the LPS challenge was observed in the CCR2 knockout mice when compared with the unchallenged group (Figure 1D and Figure 4A,B). Other chemokine receptors, such as CCR1, CCR4, and CCR5 and their corresponding ligands, may participate in the accumulation of macrophages in the absence of CCR2 [49,50]. Besides in recruitment, these ligands have a role in activation, differentiation and polarization of macrophages in numerous diseases and contexts [51]. In addition to CC chemokines and their receptors, the CX3CL1-CX3CR1 axis is also an important pathway mediating monocyte migration, playing a major, but environment-specific, role in either pro-inflammatory or pro-resolving responses [52], and contributing to the development of inflammatory diseases, such as kidney ischemia–reperfusion injury [53] and pulmonary fibrosis [50]. 

CCR2 is mainly expressed in circulating peripheral blood monocytes, but not in alveolar macrophages. It is known that alveolar macrophages originate from fetal liver monocytes and are independent of circulating monocytes [54,55]; therefore, the deficiency of CCR2 or the inhibition of CCL2 have little or no effect on this cell population [56]. Contrastingly, the interstitial macrophages originate from yolk sac progenitors and in adulthood they are replaced by circulating monocytes [57,58], thus being susceptible to CCR2 deficiency. Alveolar macrophages are crucial for the recognition and clearance of pathogens from the airways, promoting the initiation of host defense as well as the tissue repair [59]. This cell population is very important for the resolution of lung injury since they can clear apoptotic neutrophils and tissue debris through efferocytosis [60], avoiding dying cells from releasing pro-inflammatory and toxic mediators into the surroundings while activating pro-resolving and repair factors [19]. Indeed, depletion of AMs by intranasal delivery of clodronate liposomes prolonged the inflammation with higher number of leukocytes in the BALF, lack of bodyweight recovery, and worse pulmonary edema (Figure 7). Likewise, other studies already showed that depletion of alveolar macrophage in LPS-induced ALI/ARDS leads to increased influx of polymorphonuclear leukocytes [61] and more severe disease and lung inflammation [43]. Interestingly, our results show that these phenomena are more pronounced in the absence of CCR2, supporting our hypothesis that alveolar macrophages are the key cell in the control of inflammation in CCR2-deficient mice. 

According to Mahida et al. [62], ARDS in humans may be associated with impaired efferocytosis by alveolar macrophages, demonstrating how important this type of macrophage is. Our findings indeed suggest that increased proliferation of alveolar macrophages can compensate the lack of macrophages derived from monocytes, promoting proper resolution of ARDS in the absence of CCR2. It is not totally clear what causes the proliferation of alveolar macrophages in our study. Granulocyte macrophage-colony stimulating factor (GM-CSF) and macrophage-colony stimulating factor (M-CSF) are important growth factors for the expansion of alveolar macrophages [63]. Although there was no difference in GM-CSF levels between CCR2^+/+^ and CCR2^−/−^ mice at any time point evaluated, we observed a mild increase in M-CSF 3 days after the LPS instillation in the CCR2^−/−^ mice (Appendix A). M-CSF is linked with homeostasis of macrophage and monocyte populations and is able to prone monocytes towards an M2 profile, as shown by Hamilton et al. [64,65]. It must also be noted that in the CCR2^−/−^ mice, relatively more growth factor is available per target cell, as less monocytes/macrophages are present in the lungs of those animals. 

In conclusion, in our murine model CCR2 is not essential for the development, nor the resolution of ARDS induced by LPS. We observed different patterns and intensity of cell recruitment, especially in the initial phases of the inflammation, although disease development was not affected. Despite the importance of CCR2 in monocyte recruitment and the crucial role of macrophages in resolution of inflammation, our data did also not show major effects on resolution when this receptor was absent. We hypothesize that the lack of monocyte recruitment is counterbalanced by the recruitment of neutrophils, in the first days, and later by the proliferation of alveolar macrophages. More studies are necessary to further elucidate the mechanisms involved in this process and to clarify the mediators responsible for the enhanced proliferation of alveolar macrophages.

## 4. Materials and Methods

### 4.1. Mice

Eight to ten weeks old CCR2^−/−^ and CCR2^+/+^ were bred in the animal facility in the Rega Institute for Medical Research, KU Leuven. Previously, CCR2^−/−^ mice were bought from The Jackson Laboratory (B6.129S4-Ccr2tm1Ifc/J; #004999; Bar Harbor, ME, USA) and CCR2^+/+^ C57BL/6J mice from Charles River (JAX™ C57BL/6J SOPF Mice; #680; Ecully, France). Knockout and wild type mice were mated to generate F1 heterozygotes that were inter-crossed to create littermates. All animals were maintained with ad libitum water and food (Ssniff Spezialdiäte, Soest, Germany), in a 12 h dark–light cycle and kept in a controlled environment. All the experiments were performed within the norms of the European Union (directive 2010/63/EU) and the Belgian Royal Decree of 29/05/13. They were approved by the Animal Ethics Committees of KU Leuven (P101/2020) and UFMG (420/2018).

### 4.2. ARDS Model

To induce ARDS, 30 µL of *Escherichia coli* LPS (Sigma-Aldrich, Saint-Louis, MO, USA, 12.5 μg/mouse) was administered intranasally to CCR2^−/−^ and CCR2^+/+^ mice. Control animals received the same amount of endotoxin-free phosphate-buffered saline (PBS, Lonza, Walkersville, MD, USA). Body weight was measured daily, and the mice were euthanized at different time points after the instillation (1, 2, 3, 4 or 5 days). For the dissection, mice received an intraperitoneal (i.p.) injection of 100 μL of dolethal (Vetoquinol, Niel, Belgium; 200 mg/mL). Broncho-alveolar lavage fluid (BALF) was obtained by the instillation of 500 μL of PBS through a catheter in the trachea. The fluid was withdrawn and instilled again two more times, PBS instillation was repeated three times, and the lavages were pooled. After perfusion with PBS, lungs were collected for analysis by flow cytometry. The BALF was centrifuged (5 min, 300× *g*, 4 °C) and the supernatant was collected for the analysis of the cytokine levels by ELISA and protein levels by BCA, whereas the cell pellet was combined with the cells isolated from the lungs for flow cytometry analysis.

### 4.3. BALF Protein Concentration

To assess the edema formation and the extend of the tissue damage, the concentration of protein in the BALF was measured using the Pierce BCA protein assay (ThermoFisher, Waltham, MA, USA). Briefly, this assay comprises mixing the BCA working reagent with protein standards and samples followed by an incubation at 37 °C for 30 min. The microplate is cooled to room temperature and the absorbance is read at 562 nm.

### 4.4. Isolation of Single Cells from the Lungs

During dissection, lungs were removed, cut in small pieces, and collected in RPMI medium [RPMI GlutaMAX (ThermoFisher) + 5% FCS + 1% penicillin/streptomycin (ThermoFisher)] at room temperature (RT). Lungs were then incubated for 30 min at 37 °C in RPMI medium with digestive enzymes [2 mg/mL collagenase D (Sigma-Aldrich) and 0.1 mg/mL DNase I (Sigma-Aldrich)]. The tissue was homogenized using a needle and syringe and fresh digestion medium was added for a second incubation at 37 °C for 15 min. After a second process of homogenization, the samples were centrifuged (5 min, 400× *g*, RT), and the pellet was resuspended in 1 mL of 10 mM EDTA dissolved in PBS to stop the digestion. Cells were suspended in PBS + 2% FCS, centrifuged again, and treated with ACK lysing buffer (ThermoFisher) to lyse RBCs. Subsequently, they were passed through a 70 µm cell strainer and resuspended in PBS + 2% FCS. To determine the number of live cells per mL, they were diluted in trypan blue solution and counted using a Bürker chamber. Cells from the lungs were combined with cells from the BALF for flow cytometry analysis.

### 4.5. Staining and Flow Cytometry

One million cells, 3 million in the case of intracellular staining, per sample were transferred to 96 well plates and washed with PBS. They were incubated for 15 min at RT in the dark with a viability dye, Zombie UV (1/1,000; BioLegend, San Diego, CA, USA), and mouse Fc blocking reagent (MACS Miltenyi Biotec, Bergisch Gladbach, Germany). After the incubation time, the cells were washed with FACS buffer (PBS + 2% FCS + 2 mM EDTA) and stained with different panels of monoclonal antibodies (Appendix A) diluted in brilliant stain buffer (BD Biosciences; Erembodegem, Belgium) for 20 min at 4 °C in the dark. The samples were washed with FACS buffer, fixed in 0.4% formaldehyde in PBS, and transferred to FACS tubes. For the intracellular staining, the surface staining was performed and, instead of using formaldehyde, they were submitted to fixation and permeabilization using the fix/perm reagent (eBioscience, San Diego, CA, USA) for 45 min at RT in the dark, washed with the permeabilization buffer (eBioscience), incubated with the antibodies binding intracellular antigens (Appendix A) for 30 min at RT in the dark, and washed again with permeabilization buffer (eBioscience). The samples were analyzed with a BD LSR Fortessa Flow cytometer (BD Biosciences) and 100,000 live single cells were acquired. For the analysis of the data, FlowJo V10 software (BD Biosciences) was used, and the gating strategies are described in the Appendix A (Appendix A).

### 4.6. Proliferation Assays

#### 4.6.1. Ki-67 Staining

Ki-67 is a nuclear protein expressed by proliferating cells and very often used as a proliferation marker. After the isolation of single cells from the lungs, 3 million cells per sample were transferred to 96 well plates and the intracellular staining was performed as described above with the antibodies described in the Appendix A. 

#### 4.6.2. BrdU Staining

Another method to evaluate the cell proliferation is the use of 5-bromo-2’-deoxyuridine (BrdU – Sigma-Aldrich). One day before the euthanasia, wild type and knockout mice received an i.p. injection of BrdU (1.5 mg/mouse). After the euthanasia and tissue processing, flow cytometry staining was performed as aforementioned. For the intracellular staining, the cells were permeabilized two extra times and treated with DNAse to expose incorporated BrdU before the staining with the anti-BrdU antibody (Appendix A).

### 4.7. Quantitation of Neutrophil Products, Growth Factors and Cytokines in BALF by ELISA

Aliquots of cell free BALF were used for the analysis of TNF-α, IFN-γ, GM-CSF, M-CSF, NGAL, CCL2, CCL22, and CXCL1 by ELISA according to the manufacturer’s instructions (R&D Systems, Abingdon, UK). Absorbance was measured at 450 nm using a Biotek photometer (Shoreline, WA, USA) and the Gen5 software (version 2.09, Biotek, Shoreline, WA, USA).

### 4.8. Histology

Lungs for histopathological analysis were collected and inflated via the trachea with 4% formaldehyde (Sigma-Aldrich) in PBS. The samples were fixed overnight using the same solution, processed with different concentrations of ethanol and xylol, embedded in paraffin, and sectioned (5 μm). Sections were stained with hematoxylin and eosin for the evaluation of the intensity and extension of polymorphonuclear infiltrates in different lung compartments, characterizing airway inflammation, vascular inflammation, and parenchymal inflammation, as described by Horvat et al. [66]. According with the histopathological score, the tissue damage was classified as absent, mild, moderate, intense, and severe. The analysis was performed by an independent pathologist that was blinded to the experimental conditions.

### 4.9. qPCR Analysis

Following dissection, small lungs were removed from the mice and stored on dry ice until further use. Using the Qiagen RNeasy mini kit (cat #74106; Qiagen, Germantown, MD, USA), the lungs were subjected to homogenization and RNA extraction according to the manufacturer’s instructions. Subsequently, the RNA was converted to cDNA using the high-capacity cDNA Reverse Transcriptase kit (cat #4368814; Applied Biosystems, San Francisco, CA, USA). IDT primers were used to analyze the gene expression of *Siglec5* (Mm.PT.58.6685529), *Mrc1* (Mm.PT.58.42560062), *Nos2* (Mm.PT.58.43705194), *Arg1* (Mm.PT.58.8651372), *IL-1b* (MM.PT.58.42940223) and *IL-12b* (Mm.PT.58.12409997). *Ppia* (Mm.PT.39a.2.gs) was used as the housekeeping gene. Per reaction, 10 ng cDNA was used. qPCR was performed using the TaqMan Gene Expression Mastermix (cat #4369016, Applied Biosystems) and the 7500 Real-Time PCR system (Applied Biosystems). Relative gene expression was determined using the 2^−ΔΔCt^ method. 

### 4.10. Depletion of Alveolar Macrophages Using Clodronate Loaded Liposomes

For depletion of alveolar macrophages, 0.5 mg of clodronate in 100 µL (Liposoma, Amsterdam, The Netherlands) was intranasally instilled in mice under anesthesia 48 and 24 h before the LPS challenge. The same volume of PBS-loaded liposomes was instilled in the control groups [67]. Four days after the LPS challenge, mice were euthanized, and the dissection was conducted as described in the topic 4.2.

### 4.11. Statistics

The data were analyzed using the GraphPad PRISM software (version 9.0.0, GraphPad, La Jolla, CA, USA,). The data was checked for normality by Shapiro–Wilk test and Kolmogorov–Smirnov test. The data with normal distribution were submitted to the one-way ANOVA test followed by the Bonferroni correction. In case normality was not observed, Kruskal–Wallis with Dunn’s multiple comparisons test was performed. If only two groups were to be compared, Mann–Whitney U test was performed. Significance was determined between each condition for the CCR2^+/+^ and for the CCR2^−/−^ mice and between the CCR2^+/+^ and CCR2^−/−^ mice within each condition. Statistical differences are indicated with an asterisk above the individual data sets when compared to the corresponding control group and with horizontal lines with hashtag on top in case of comparison between the indicated wild-type and knockout groups. *p*-values were indicated as follows: * = *p* < 0.05 when compared to the control group and # = *p* < 0.05 when comparing wild-type and knockout groups.

## Figures and Tables

**Figure 1 ijms-23-12920-f001:**
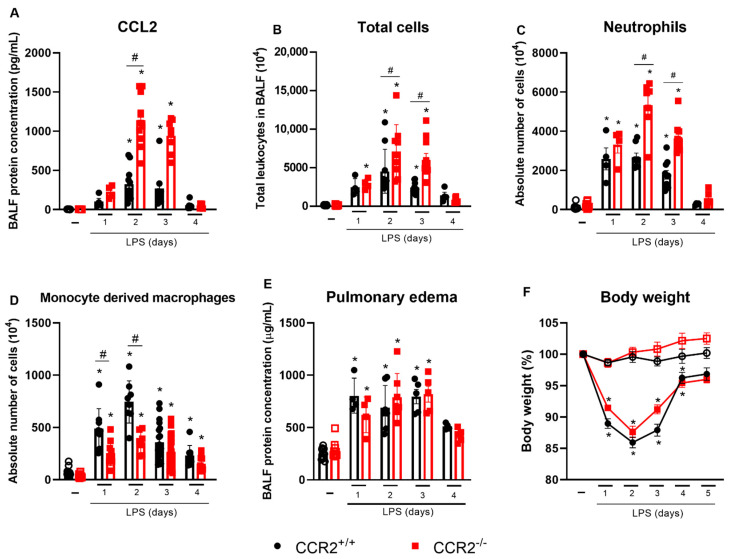
CCR2 absence results in increased accumulation of neutrophils and decreased macrophage numbers in the lungs without affecting changes in inflammation, pulmonary edema, or weight loss. CCR2^+/+^ (black symbols) and CCR2^−/−^ (red symbols) C57BL/6 mice were challenged with LPS (12.5 µg/mouse) or PBS (ctrl group; -) and dissected at the indicated days. Levels of CCL2 (**A**) were measured in the BALF by ELISA. Absolute numbers of leukocytes in BALF (**B**) were counted in Bürker chamber. Absolute numbers of neutrophils (CD45^+^Ly6G^+^CD11b^+^) (**C**) or macrophages (CD45^+^CD11b^+^Ly6G^−^CD3^−^CD19^−^ NKp46^−^CD103^−^SiglecF^−^CD11c^−^ cells) (**D**) isolated from the lungs and BALF were quantified by flow cytometry. Pulmonary edema was quantified based on the protein concentration in the BALF (**E**). Changes in body weight (**F**) were calculated with the weight before challenge (day 0) as reference. Compilation of three experiments. Data are shown as mean ± SEM. Each symbol in panels A to E represents data of an individual mouse. * *p* < 0.05 when compared with the healthy, unchallenged control group. # *p* < 0.05 when comparing wild type and knockout group at the same time point. ANOVA test followed by Bonferroni correction was used in panel F; Kruskal–Wallis with Dunn’s multiple comparisons test was used in panels A–E. *n* = 6–12.

**Figure 2 ijms-23-12920-f002:**
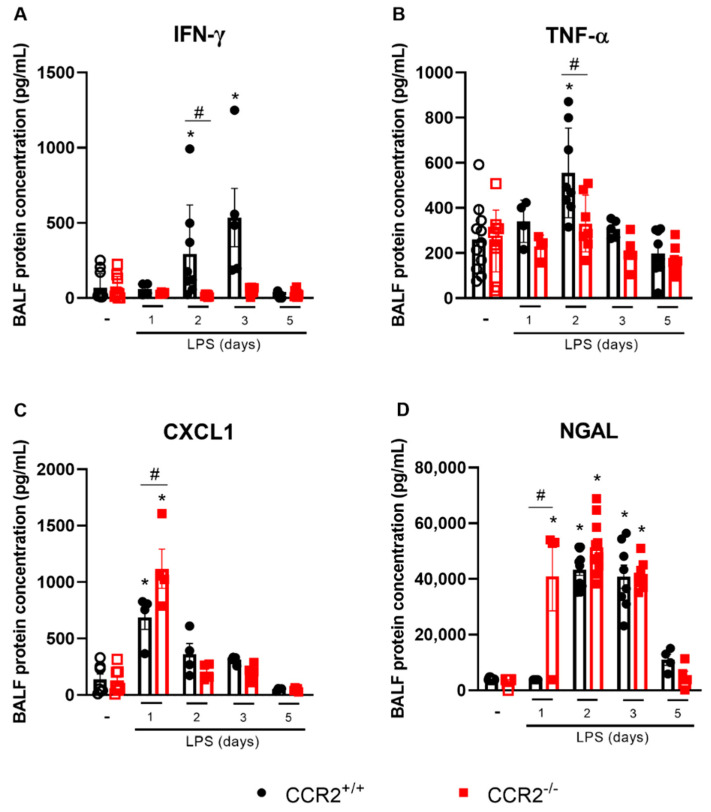
CCR2 deficiency affects cytokine levels in the pro-inflammatory phase of the inflammation. CCR2^+/+^ (black symbols) and CCR2^−/−^ (red symbols) C57BL/6 mice were challenged with LPS (12.5 µg/mouse) or PBS (ctrl group; -) intranasally and dissected at the indicated days. Levels of IFN-γ (**A**), TNF-α (**B**), CXCL1 (**C**), and NGAL (**D**) were measured in the BALF by ELISA. Compilation of three experiments. Data are shown as mean ± SEM. Each symbol represents data of an individual mouse. * *p* < 0.05 when compared with the healthy, unchallenged control group. # *p* < 0.05 when comparing wild type and knockout group at the same time point. ANOVA test followed by Bonferroni correction was used in panel B; Kruskal–Wallis with Dunn’s multiple comparisons test was used in panels A, C, and D. *n* = 4–12.

**Figure 3 ijms-23-12920-f003:**
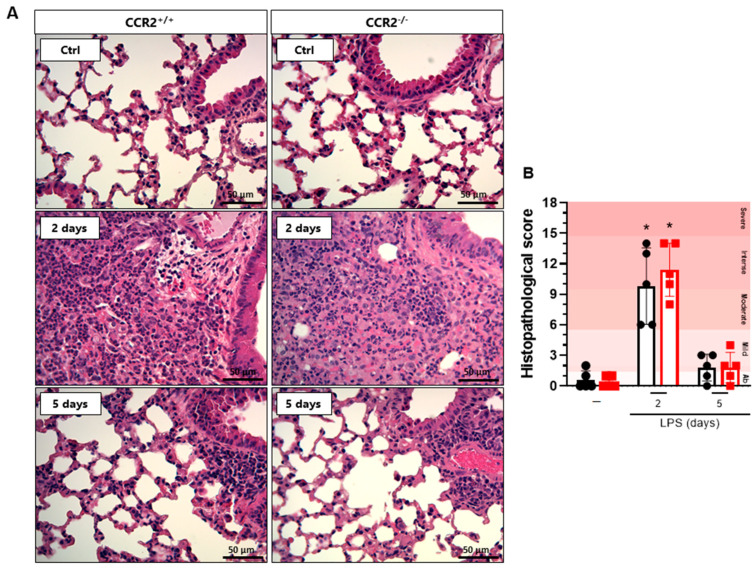
CCR2-deficiency does not influence the histopathological score in CCR2^−/−^ compared to CCR2^+/+^ mice. CCR2^+/+^ (black symbols) and CCR2^−/−^ (red symbols) C57BL/6 mice were challenged with LPS (12.5 µg/mouse) or PBS (ctrl group; -) and dissected after 2 or 5 days. (**A**) Representative hematoxylin and eosin-stained preparations of lung tissue from mice. Scale bar: 50 μm, as reported in the figure. (**B**) Histopathological score with ranges of tissue damage (severe, intense, moderate, mild, or absent). Data are shown as mean ± SEM from one representative out of two independent experiments. Each symbol represents data of an individual mouse. * *p* < 0.05 when compared with the healthy, unchallenged control group (Kruskal–Wallis with Dunn’s multiple comparisons test). *n* = 5.

**Figure 4 ijms-23-12920-f004:**
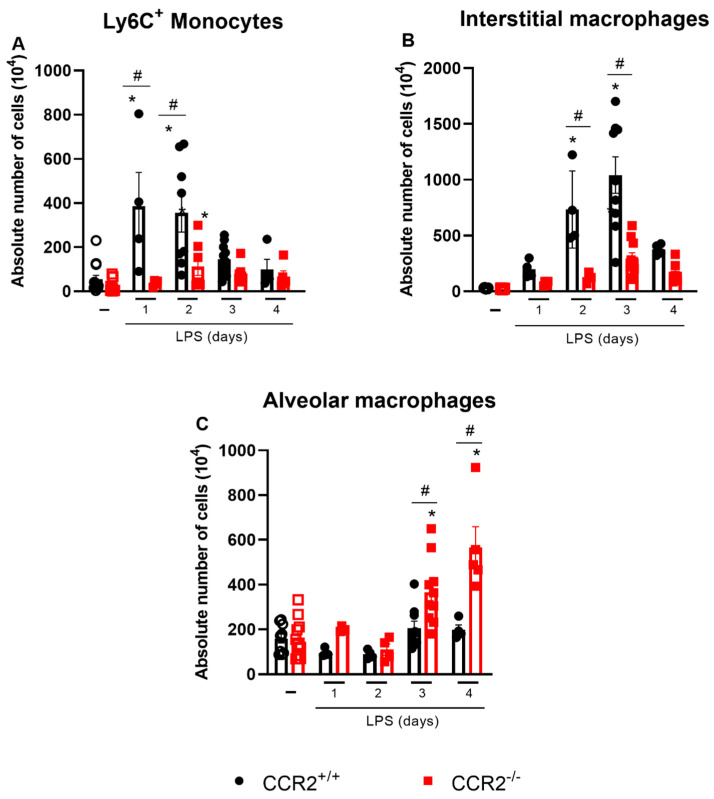
Largely reduced numbers of Ly6C^+^ monocytes and interstitial macrophages but increased alveolar macrophage counts are observed in CCR2^−/−^ compared to CCR2^+/+^ mice. CCR2^+/+^ (black symbols) and CCR2^−/−^ (red symbols) C57BL/6 mice were challenged with LPS (12.5 µg/mouse) or PBS (ctrl group; -) and dissected at the indicated days. Absolute numbers of Ly6C^+^ monocytes (CD45^+^CD11b^+^SiglecF^−^Ly6G^−^Dump^−^CD103^−^MHCII^−^Ly6C^+^ cells) (**A**) interstitial macrophages (CD45^+^CD11b^+^SiglecF^−^Ly6G^−^Dump^−^CD103^−^MHCII^+^ cells) (**B**) and alveolar macrophages (CD45^+^SiglecF^+^CD11c^+^ cells) (**C**) were quantified by flow cytometry. Compilation of three experiments. Data are shown as mean ± SEM. Each symbol represents data of an individual mouse. * *p* < 0.05 when compared with the healthy, unchallenged control group. # *p* < 0.05 when comparing wild type and knockout group at the same time point. ANOVA test followed by Bonferroni correction was used in panels B and C; Kruskal–Wallis with Dunn’s multiple comparisons test was used in panel A. *n* = 4–12.

**Figure 5 ijms-23-12920-f005:**
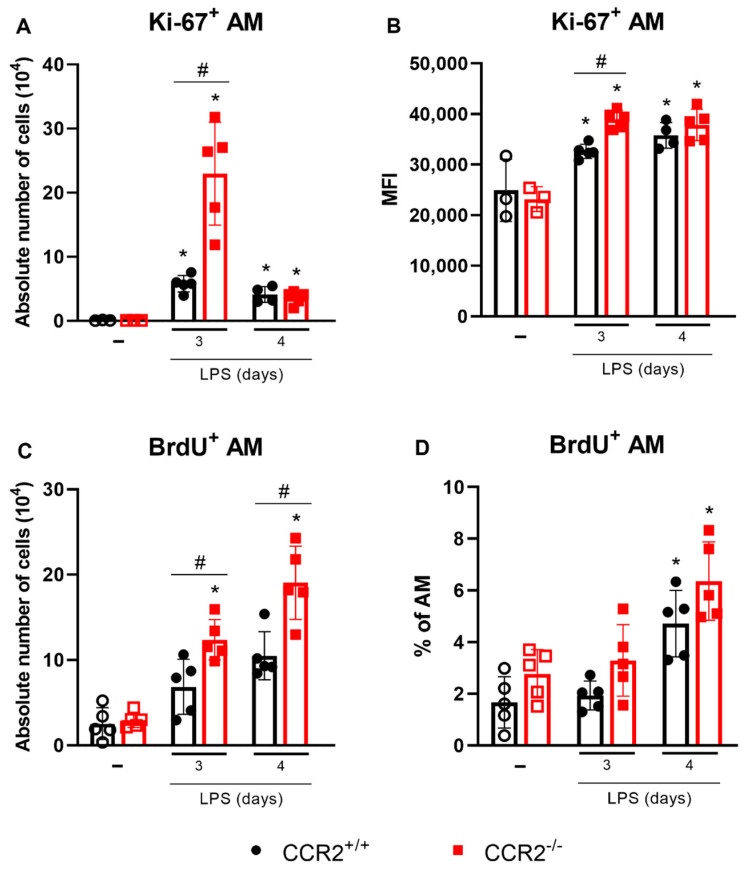
CCR2^−/−^ mice show increased proliferation of alveolar macrophages. CCR2^+/+^ (black symbols) and CCR2^−/−^ (red symbols) C57BL/6 mice were challenged with LPS (12.5 µg/mouse) or PBS (ctrl group; -) and dissected at the indicated days. (**A**) Absolute number of alveolar macrophages expressing Ki-67 quantified by flow cytometry using the following markers: CD45^+^CD11c^+^SiglecF^+^Ki-67^+^. (**B**) Mean fluorescence intensity (MFI) of Ki-67 in alveolar macrophages. (**C**) Absolute number of BrdU^+^ alveolar macrophages quantified by flow cytometry using the following markers: CD45^+^CD11c^+^SiglecF^+^BrdU^+^. (**D**) Percentage of BrdU^+^ alveolar macrophages. Data are shown as mean ± SEM from one representative out of two independent experiments. Each symbol represents data of an individual mouse. * *p* < 0.05 when compared with the healthy, unchallenged control group (ANOVA test followed by Bonferroni correction). # *p* < 0.05 when comparing wild type and knockout group at the same time point (ANOVA test followed by Bonferroni correction). *n* = 3–5.

**Figure 6 ijms-23-12920-f006:**
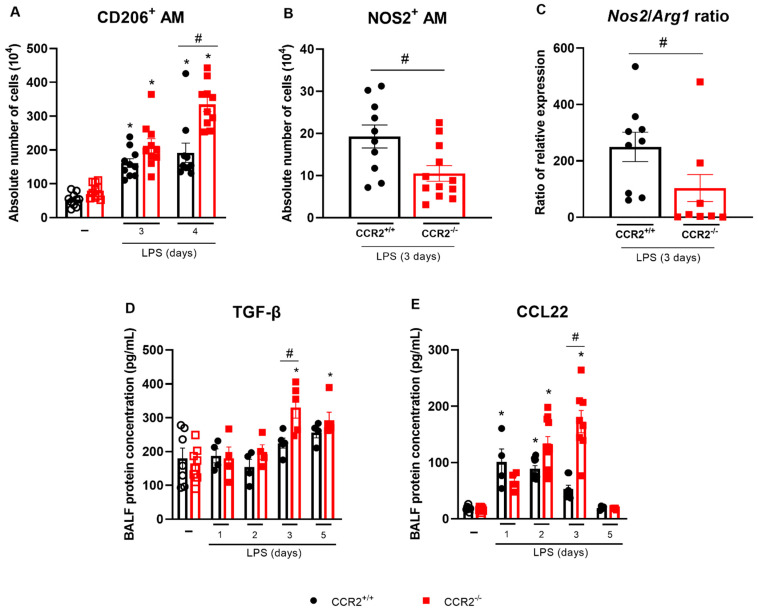
CCR2 deficiency is associated with the increase of molecules related with M2 macrophages. CCR2^+/+^ (black symbols) and CCR2^−/−^ (red symbols) C57BL/6 mice were challenged with LPS (12.5 µg/mouse) or PBS (ctrl group; -) and dissected at the indicated days. Absolute numbers of CD206^+^ alveolar macrophages (CD45^+^CD11c^+^SiglecF^+^CD206^+^ cells) (**A**) and NOS2^+^ alveolar macrophages (CD45^+^CD11c^+^SiglecF^+^NOS2^+^ cells) (**B**) quantified by flow cytometry. (**C**) Ratio of *Nos2* and *Arg1* mRNA expression relative to the endogenous control. Levels of TGF-β (**D**) and CCL22 (**E**) in BALF quantified by ELISA. Compilation of three experiments in panels A, D, and E; and two experiments in panels B and C. Data are shown as mean ± SEM. Each symbol represents data of an individual mouse. * *p* < 0.05 when compared with the healthy, unchallenged control group. # *p* < 0.05 when comparing wild type and knockout group at the same time point. ANOVA test followed by Bonferroni correction was used in panels A and E; Kruskal–Wallis with Dunn’s multiple comparisons test was used in panel D. Mann–Whitney U test was used in panels B and C. *n* = 4–12.

**Figure 7 ijms-23-12920-f007:**
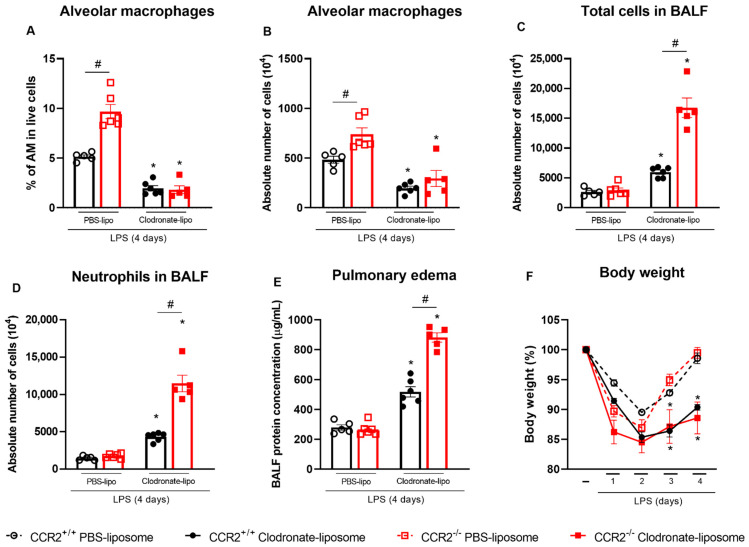
Depletion of AM leads to worsened inflammation especially in CCR2^−/−^ mice. CCR2^+/+^ (black symbols) and CCR2^−/−^ (red symbols) C57BL/6 mice were treated intranasally with liposomes loaded with clodronate or PBS. One day later, they were challenged with LPS (12.5 µg/mouse) and after 4 days mice were euthanized. Percentage (**A**) and absolute numbers (**B**) of alveolar macrophages (CD45^+^CD11c^+^SiglecF^+^ cells) isolated from the lungs and BALF were quantified by flow cytometry. Total leukocytes (**C**) and neutrophils (**D**) in BALF were counted microscopically. Pulmonary edema was quantified based on the protein concentration in the BALF (**E**). Changes in body weight (**F**) were calculated with the weight before challenge (day 0) as reference. Data are shown as mean ± SEM. Each symbol represents data of an individual mouse. * *p* < 0.05 when compared with the respective group treated with PBS-loaded liposomes. # *p* < 0.05 when comparing wild type and knockout group treated with clodronate-loaded liposomes. Mann–Whitney U test was used. *n* = 5–6.

## Data Availability

All data generated in this study are included in the manuscript.

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
