# Peer review of "Absence of CCR2 Promotes Proliferation of Alveolar Macrophages That Control Lung Inflammation in Acute Respiratory Distress Syndrome in Mice"

_ijms, 2022, doi:10.3390/ijms232112920_

Round 1

Reviewer 1 Report

This is a technically sound study, and as such it may be published. However, the phenotypes described are only weak to moderate and were obtained in a rapid, self-curing mouse model with ~48h phenotype duration which make the conclusions a bit vague. In addition, I was unable to download Supplementary materials but this does not influence much my opinion. I recommend the following changes to the authors and express some concerns. 

Methodology:

1. Why lung histopathology was evaluated by scoring rather than by densitometry, which formally is more objective?

2. Looks like evaluation of apoptosis in neutrophils would be more precise if triple Annexin V/7AA/Ly6G staining with FACS analysis would be applied instead of morphology.

Figures:

Fig. 1., and onwards: It is difficult to follow comparisons between KO vs. wild type groups + LPS-treated vs. non-treated animals in a single-panel format. I think that the authors might consider transformation of at least some figures into dynamic curves comparing only one, more important, parameter (KO vs. wild type), with only limited presentation (Suppl. material?) of another. An additional complication is a very small scale preventing a clear look at individual dots. An option for some enlargement here is to remove day 5 from the graphs since it adds almost nothing to the general picture. In addition, it is not appropriate to display graphs with multiple asterisks indicating significance (as on Fig. 1E and F), whilst indicating “no difference” in the text. The reader should be supplied with as clear illustrations as possible. In this article, it is not always clear which groups are compared.

Fig. 3 and its description may be moved to Supplementary materials – it does not lead to any important conclusion.

Fig. 6. The figures for absolute numbers of apoptotic neutrophils in KO and w.t. mice show that there is no significant differences at day 2, but significance appears at day 3. This looks odd, given that means look more distant whereas deviations bigger at day 3.

Evaluation of iNOS/Arginase production and ratios are desirable for macrophage polarization assessment – a single measurement of the CD206 marker looks insufficient.

I recommend to cut the Discussion section a bit, or make it more imaginative to rise the interst of the readers.

Reviewer 2 Report

In this study, the authors precisely observed that three independent components of immune cells (monocyte-derived macrophages, neutrophils, and alveolar macrophages) are mutually regulated during a sequential process of acute respiratory distress syndrome (ARDS) using CCR2 KO mouse. Each observation of immune cell population by flowcytometry is quite accurate and reliable. However, it seems to be a just lineup of alterations in immune cell composition. Thus, the following several points should be addressed to clarify their causal relationships in this pathology.

 1. The authors should do the immunostaining or flowcytometric analysis to identify the major cellular source of CXCL1 in the lung tissues.

 2. To directly examine the effect of CCR2-deficiency-mediated reduction of migratory macrophages on the compensatory proliferation of alveolar macrophages, depletion of circulating monocytes prior to the development of ARDS model should be done by intravenous injection of clodronate liposome or Macrokiller (commercially available).

 3. Similarly, the authors are strongly suggested to deplete circulating granulocytes and alveolar macrophages by intravenous injection of Ly6G mAb and intratracheal injection of clodronate liposome, respectively, to directly examine their effect on the pathology of ARDS.

Round 2

Reviewer 2 Report

I suppose that the authors have answered most of my questions sufficiently.